# Platinum-nickel alloy excavated nano-multipods with hexagonal close-packed structure and superior activity towards hydrogen evolution reaction

Zhenming Cao[1], Qiaoli Chen[1], Jiawei Zhang[1], Huiqi Li[1], Yaqi Jiang[1], Shouyu Shen[1], Gang Fu[1], Bang-an Lu[1], Zhaoxiong Xie[1,2] & Lansun Zheng[1]

Crystal phase regulations may endow materials with enhanced or new functionalities. However, syntheses of noble metal-based allomorphic nanomaterials are extremely difficult, and only a few successful examples have been found. Herein, we report the discovery of hexagonal close-packed Pt–Ni alloy, despite the fact that Pt–Ni alloys are typically crystallized in face-centred cubic structures. The hexagonal close-packed Pt–Ni alloy nano-multipods are synthesized via a facile one-pot solvothermal route, where the branches of nano-multipods take the shape of excavated hexagonal prisms assembled by six nanosheets of 2.5 nm thickness. The hexagonal close-packed Pt–Ni excavated nano-multipods exhibit superior catalytic property towards the hydrogen evolution reaction in alkaline electrolyte. The overpotential is only 65 mV versus reversible hydrogen electrode at a current density of 10 mA cm$^{-2}$, and the mass current density reaches 3.03 mA μgPt$^{-1}$ at −70 mV versus reversible hydrogen electrode, which outperforms currently reported catalysts to the best of our knowledge.

[1] State Key Laboratory of Physical Chemistry of Solid Surfaces, Collaborative Innovation Center of Chemistry for Energy Materials, and Department of Chemistry, College of Chemistry and Chemical Engineering, Xiamen University, Xiamen 361005, China. [2] Pen-Tung Sah Institute of Micro-Nano Science and Technology, Xiamen University, Xiamen 361005, China. Correspondence and requests for materials should be addressed to Y.J. (email: yqjiang@xmu.edu.cn) or to G.F. (email: gfu@xmu.edu.cn) or to Z.X. (email: zxxie@xmu.edu.cn).

Noble metal-based nanomaterials are promising functional materials, which have potential applications in many important fields. The discovery of new crystal phases provides an opportunity for achieving novel functionalities due to different atomic arrangements and electronic structures of allomorphs, and thus is an important approach for new materials/catalysts development[1–3]. Although allomorphs of noble metal-based nanocrystals (NCs) have shown fascinating properties in optics, magnetism and catalysis, their crystal phase regulations are extremely difficult[4–6]. So far, only a few meta-stable phases of noble metal-based NCs have been reported, and most of them are achieved in harsh conditions (such as high temperatures[7,8] and pressures[9,10]) or with a specific template[11–13]. Syntheses of noble metal NCs with meta-stable crystal phases remain challenging, especially in mild synthetic conditions.

Benefiting from electronic and synergistic effects, Pt–Ni alloy is one of the best catalysts for many energy conversion applications. For example, Pt–Ni alloy can effectively reduce the overpotential and improve the sluggish kinetics of electrocatalytic oxygen reduction and hydrogen evolution (HER)[14–16]. In the past few decades, great efforts have been devoted to improving the catalytic efficiency through tailoring the size, morphology and composition of Pt–Ni alloy NCs[17–19]. However, all the reported Pt–Ni alloys adopt the typical face-centred cubic (fcc) crystal phase, failing to produce NCs in meta-stable crystal phases.

Here we report the successful synthesis of hexagonal closed-packed (hcp) Pt–Ni alloy nano-multipods in a mild solvothermal condition. Each branch of Pt–Ni nano-multipods takes the shape of an excavated hexagonal prism assembled by six ultrathin nanosheets of 2.5 nm thickness, endowing these nano-multipods with a large surface area. It is demonstrated that the unexpected hcp Pt–Ni alloy nanostructure exhibits superior catalytic property towards HER. Particularly, to the best of our knowledge, the mass current density of hcp Pt–Ni excavated nano-multipods exhibits the highest value among currently reported Pt-based catalysts.

## Results

**Crystal structure and composition characterization.** Figure 1a depicts the representative scanning electron microscopy (SEM) image of the sample prepared by co-reducing metal precursors $Pt(acac)_2$ and $Ni(acac)_2$ in a solvothermal condition. It can be seen that the product is composed of nano-multipods (see also Supplementary Fig. 1), and the average length and diameter of the branches are about 145 nm and 35 nm, respectively. Crystal structure information was acquired by powder X-ray diffraction (XRD), which shows totally different diffraction pattern from the common fcc structured Pt-based alloy. The diffraction peaks can be indexed to a hcp phase with a unit cell of $a = b = 2.6367(1)$ Å and $c = 4.3310(2)$ Å, as shown in Fig. 1b. Further structural information was acquired by transmission electron microscopy (TEM) and the corresponding selected area electron diffraction (SAED) of single branches, as shown in Fig. 1c,e, respectively. The high-resolution TEM image of the single branch (Fig. 1d) shows that the atom packing mode is ABABAB, which is in accordance with the unique hcp crystal phase. The corresponding SAED pattern (Fig. 1e) can be indexed as diffractions along [1$\bar{1}$00] zone axis, which indicates that single branches grow along the $<0001>$ direction of hcp crystal phase. This directional growth is also consistent with the sharp diffraction peak of (0002) in the XRD pattern, whose full-width at half-maxima is smaller than that of the others.

To confirm the composition of the as-prepared product, energy dispersive X-ray spectroscopy (EDS) and inductively coupled

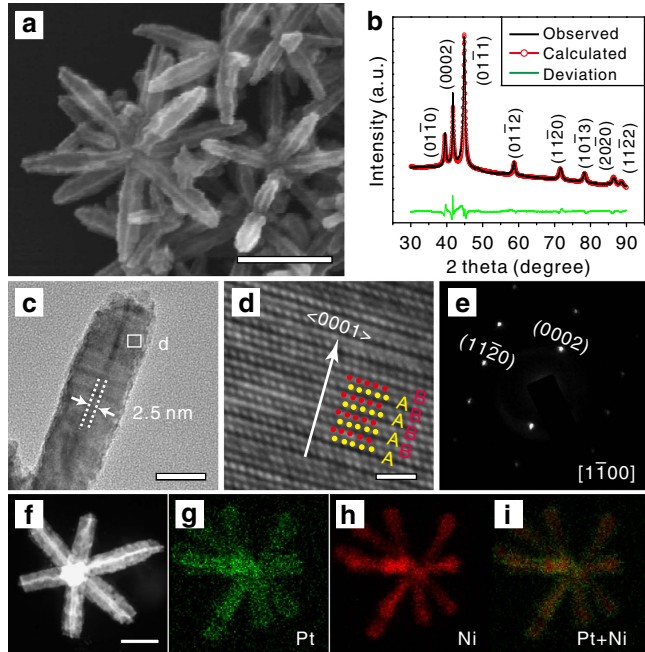

**Figure 1 | Structure and composition analysis of hcp Pt–Ni excavated nano-multipods.** (**a**) SEM image (scale bar, 150 nm), and (**b**) observed XRD pattern, calculated XRD pattern (by Rietveld refinement using an hcp Pt–Ni structure model), and their deviations. (**c**) TEM image (scale bar, 20 nm), (**d**) high-resolution TEM image (scale bar, 1 nm), and (**e**) the corresponding SAED pattern taken along [1$\bar{1}$00] direction of a single branch. (**f**) HAADF-STEM image (scale bar, 50 nm) and (**g–i**) EDS mapping of Pt and Ni elements of the excavated Pt–Ni nano-multipods.

plasma atomic emission spectrometry (ICP-AES) analyses were carried out. EDS analysis shows the as-prepared product is consistent with Pt–Ni alloys with Pt content of 12 at% (Supplementary Figs 2 and 3). The ICP-AES measurement shows a total atomic concentration of Pt of 11.5 at%, which is in good agreement with the EDS result. Furthermore, high-angle annular dark field scanning transmission electron microscopy (HAADF-STEM) and EDS mapping (Fig. 1f–i) shows that both Pt and Ni atoms uniformly distribute over the whole branched nanostructure, confirming the Pt and Ni are well alloyed. Taking the composition and the alloying nature into consideration, Rietveld refinement based on the full XRD pattern was then conducted by applying the hcp Pt–Ni alloy structure model. As shown in Fig. 1b and Supplementary Table 1 and Note 1, the calculated XRD pattern agrees well with the measured one, showing low values of reliability factors, $R_p$ and $R_{wp}$ (2.05 and 2.97, respectively), which further supports that the Pt–Ni alloy is in the hcp phase.

**Characterization of excavated polyhedral feature.** The SEM, TEM and HAADF-STEM images (Fig. 1) show that each branch of the nano-multipods is an excavated hexagonal prism assembled by six ultrathin nanosheets. The thickness of each nanosheet is about 2.5 nm (Fig. 1c). The surface of the ultrathin nanosheets can be determined to be {11$\bar{2}$0} facets from the TEM image (Fig. 1c) and corresponding SAED pattern (Fig. 1e). It should be noted that the {11$\bar{2}$0} facets are of the highest surface energy among basic crystal facets (that is, {0001}, {10$\bar{1}$0} and {11$\bar{2}$0}) with only atomic terrace in the hcp structure, where each atom on the surface only coordinates with seven metal atoms (Supplementary Fig. 4).

Further detailed morphological information was explored from the TEM images and corresponding models viewed along

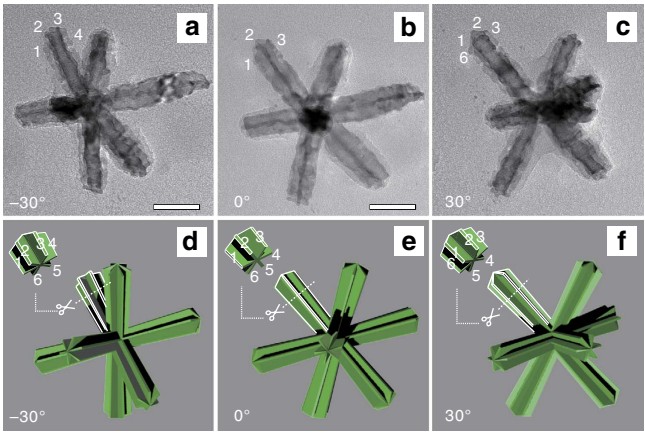

**Figure 2 | Detailed morphology analysis of the hcp Pt–Ni alloy nano-multipods.** (**a**–**c**) TEM images and (**d**–**f**) the corresponding schematic models of a single Pt-Ni excavated nano-multipod viewed from different orientations. The Arabic numbers identify the nanosheets in both the TEM images and corresponding schematic models. The tilted angles of the sample holder for (**a**–**c**) are −30°, 0° and 30°, respectively. Scale bar, 50 nm.

different angles (see Fig. 2 and Supplementary Fig. 5 for details). For clarity, we focus on the image variations of the branch in top-left along with the change of the tilt angle of the TEM sample holder. The six nanosheets are marked with number from 1 to 6, respectively, in TEM images and corresponding models of Fig. 2. The insets in Fig. 2d–f show an expanded view of the orientations of the six nanosheets at tip of the top-left branch. It can be seen the corresponding models fit the TEM images very well. When the tilt angle of TEM sample holder is set at −30°, nanosheets No. 1 and 4 align along a plane perpendicular to the electron beam (Fig. 2d), while nanosheets No. 2 and 3 locate above the plane, which overlap with nanosheets No. 5 and 6, respectively, that are below the plane, giving rise to their darker contrast in the TEM image (Fig. 2a). At 0° tilt angle, nanosheets No. 2 and 5 align vertically (parallel to the electron beam) and overlay on top of each other (Fig. 2e), appearing as the narrowest dark lines in the TEM image (Fig. 2b). With a tilt angle of 30°, nanosheets No. 1 and 2 overlap with nanosheets No. 4 and 5 (Fig. 2c,f), which is similar to the case of Fig. 2a. The good agreement between the TEM images and the models confirms that the branch of nano-multipod is excavated and assembled by six ultrathin nanosheets.

**Formation of the hcp Pt–Ni alloy phase.** For Pt, Ni and their alloys, fcc structure is the most stable phase. The hcp Pt–Ni alloy is a thermodynamically meta-stable phase. In fact, the as-prepared hcp Pt–Ni alloy nano-multipods can be transformed into fcc phase by heating at 350 °C under the protection of reducing gas (5% $H_2$ + 95% $N_2$) (Supplementary Fig. 6). To obtain a deeper insight into the formation of the hcp structure, a series of control experiments was conducted. It was found that the products were fcc Pt–Ni alloy NCs when no formaldehyde was added or the usage of formaldehyde was only 200 μl (Supplementary Figs 7 and 8). As the amount of the formaldehyde solution increased to 400 μl or more, the products changed to hcp Pt–Ni phase and their morphology changed to nano-multipods. The experiment data suggest that formaldehyde plays a crucial role in the formation of hcp Pt–Ni NCs and final morphology of excavated nano-multipods.

It is thought that formaldehyde can decompose into CO and $H_2$ in the presence of Pt NCs[20]. We thus carried out control

experiments by replacing formaldehyde with CO, $H_2$ and a mixture of CO and $H_2$, respectively, while keeping other reaction conditions unchanged. When CO was introduced, the products were $Pt_{78}Ni_{22}$ alloy (Pt-rich) NCs of fcc phase (Supplementary Fig. 9a,b). When only $H_2$ was present in the reaction system, surprisingly, almost only Ni NCs with fcc structure was formed, and the atomic percentage of Pt was merely 0.5% (Supplementary Figs 9c,d and 10). Taking standard redox potential into consideration, $Pt^{2+}$ ions should be reduced to its metallic state (Pt atoms) more easily than $Ni^{2+}$ ions. However, the presence of $H_2$ seems to decrease the reduction of $Pt^{2+}$ ions as neither Pt nanoparticles nor obvious Pt–Ni alloys were formed. When the mixture of CO and $H_2$ (volume ratio of 1:1) was introduced, the products consisted of three different Pt–Ni alloys in the fcc structure. From the measured lattice parameters and according to Vegard's law that the lattice parameter of a solid solution alloy is linearly dependent on its composition[21], the three phases were determined to be $Pt_8Ni_{92}$ (Ni-rich), PtNi and $Pt_2Ni$, respectively (Supplementary Fig. 9e,f). As a result, both CO and $H_2$ are not the key factor in the formation of hcp Pt–Ni alloy, but the presence of $H_2$ could result in Ni-rich Pt–Ni alloys. The formation mechanism of hcp phase could be due to a synergistic effect of formaldehyde and its decomposed products, although a detailed mechanism needs to be further explored.

In addition, another control experiments were carried out to demonstrate whether the content of Ni affect the formation of hcp Pt–Ni phase by changing the molar ratio between precursors $Pt(acac)_2$ and $Ni(acac)_2$. When the molar ratio was larger than 1:1 (for example, 5:1 and 2:1), the corresponding products were nanoparticles of fcc Pt–Ni alloy and the Pt content of the final product decreased with the decreasing molar ratio (Supplementary Figs 11 and 12). With the molar ratio decreasing to 1:1, the hcp Pt–Ni alloy phase appeared, accompanying the appearance of branched structure. When the molar ratio reached 1:2, hcp Pt–Ni alloy excavated nano-multipods could be synthesized with a Pt content of 14 at%. In fact, for all the hcp phase Pt–Ni alloy obtained in our experiments, the Pt content was low. The results indicate that hcp phase of Pt–Ni alloy is related to the hcp nickel phase.

**Formation of the excavated polyhedral feature.** In the past decades, intensive efforts have been devoted to the development of low-cost catalysts for large-scale applications. One approach is to control the surface structure and increase atom utilization efficiency of noble metal NCs in an effort to optimize their catalytic performance while minimizing the usage of noble metal[22–27]. The excavated polyhedral NCs, constructed by orderly assembling of ultrathin nanosheets, combine the advantages of both well-defined surfaces and large surface areas, and thus are promising candidates for efficient and low-cost catalysts[28–32]. However, formation of the excavated structure is thermodynamically unfavourable during crystal growth. To figure out how the unique Pt–Ni excavated multipods were formed, time-dependent experiments were carried out. It can be seen that octahedral Pt–Ni alloy NCs with a size about 50 nm formed after 3 h of reaction (Fig. 3a). The XRD analysis shows that the octahedral Pt–Ni NCs belong to fcc structure (Fig. 3d) and the Pt content is about 50 at% (Supplementary Fig. 13) determined by ICP-AES measurement. As the reaction time reached 7 h, trumpet-like branches of about 100 nm (Fig. 3b) grew and stuck to the surfaces of the octahedral Pt–Ni cores. The XRD pattern shows that the hcp phase appeared in the product coexisting with the fcc phase. It is reasonable to propose that octahedral cores belong to fcc phase and the branches belong to hcp phase.

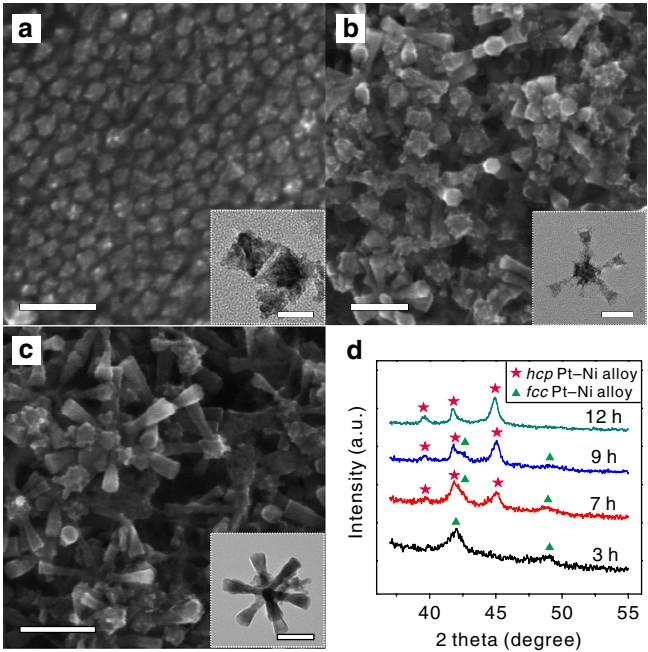

**Figure 3 | Formation of the excavated polyhedral feature of hcp Pt–Ni alloy nano-multipods.** SEM images of the Pt-Ni alloy NCs formed at reaction time of (**a**) 3 h, (**b**) 7 h and (**c**) 9 h (scale bar, 100 nm). Insets are the corresponding TEM images (scale bars, 50 (**a**), 100 (**b**) and 100 nm (**c**)). (**d**) The corresponding XRD patterns of the products.

In fact, the polymorphism existing in different parts of the nanostructure combined with the regulation of growth kinetics can facilitate the formation of branched nanostructures, as reported by Alivisatos and co-workers[33]. In our case, the atomic arrangement of {111} facets of fcc phase is identical to {0001} facets of hcp lattice. It is easy for the Ni-rich Pt–Ni alloy (hcp phase) to grow epitaxially on eight {111} facets because of low-energy barrier and form eight branches. A series of control experiments show that the branched structure can only be found when the hcp phase appears (Supplementary Figs 11a and 12), which indicates that the anisotropic growth of the nano-branched Pt–Ni alloy could be due to the anisotropic feature of the hcp structure. In addition, the selective adsorption of oleic acid on specific crystal facets may also affect the growth kinetics, as the branches cannot be formed in the absence of oleic acid (Supplementary Figs 14 and 15). Similar phenomena were found in the formation of hcp Ni, and nanorods or nanowires of CoNi alloy[34–36], where the concentration of oleic acid strongly influences the aspect ratio of their hcp nano-products. The effect of oleic acid on the anisotropic growth of multipod structure could be ascribed to their preferential adsorption on the side facets of the hcp crystal structure according to density functional theory (DFT) calculations (Supplementary Table 2 and Supplementary Note 2).

Besides the change of morphology from nanoparticle to branched structure, the total Pt content in the Pt–Ni alloy NCs dropped to 15 at%, suggesting a fast reduction process of $Ni^{2+}$ ions relative to $Pt^{2+}$ ions when forming the hcp phase. The fast reduction of $Ni^{2+}$ ions may be due to the decomposition of formaldehyde into $H_2$ and CO because the control experiment replacing formaldehyde with $H_2$ in the synthesis shows that $H_2$ accelerates the reduction of $Ni^{2+}$ ions and hinders the reduction of $Pt^{2+}$ ions (Supplementary Figs 9 and 10). In addition, DFT calculations showed that the coordination of oleylamine (the solvent) with reduced Pt atoms is much stronger than that with reduced Ni atoms, because the reduced Pt atoms have much lower-chemical potential than Ni atoms in this case (the calculation details and explanation are shown in Supplementary Table 3 and Supplementary Note 3). We thus expect that Pt atoms are preferentially reduced if a solvent with weak coordination ability is employed. Indeed, it is observed experimentally that when octadecene was used as solvent, Pt-rich Pt–Ni alloy was formed under the similar conditions (Supplementary Fig. 16), which is in accordance with the theoretical prediction.

After 9 h of reaction, the morphology of the product kept the branched structure, and the trumpet-like branches grew into solid hexagonal prisms (Fig. 3c). The atomic concentration of Pt further decreased to about 9 at% (Supplementary Fig. 11). When the reaction time reached 12 h, the ultimate product became Pt–Ni excavated nano-multipods with each branch consisting of six ultrathin nanosheets. At this stage, interestingly, the total Pt content of the product increased a little to 12 at%. Based on the fact that the formation of excavated hexagonal prisms of Pt–Ni multipods accompanied with the increase of Pt content, it is likely that either etching of Ni atoms or deposition of Pt atoms followed by reconstruction occurred at the final growth stage. A control experiment was designed to clarify the origin of the formation of excavated polyhedral structure. The solid Pt–Ni multipods (Supplementary Fig. 17a) were collected and put back to an autoclave containing 4.0 mg of $Pt(acac)_2$, then they were allowed to react for 3 h while keeping other chemical environment unchanged. It was found that all of the solid hexagonal prisms evolved to the excavated ones (Supplementary Fig. 17b). In contrast, the intermediate product kept the solid morphology if no $Pt(acac)_2$ was added in the reaction solution. The result indicated the evolution of solid hexagonal prism to excavated structure was not due to the etching mechanism. It is likely driven by preferential deposition of Pt atoms on the edge of hexagonal prisms, followed by the diffusion of Ni atoms of the Ni-rich hexagonal prism to the edge, leading to the formation of excavated Pt–Ni alloy.

The tendency of Pt deposition on the edge sites of Pt–Ni crystallites has been reported although its detailed mechanism is still not clear[15,37,38]. In our case, HAADF-STEM-EDS mapping of the product formed with 9 h reaction also shows that the Pt atoms are rich in the edge area of the branches before they evolve into excavated structure (Supplementary Fig. 18). On the other hand, it has been reported that the presence of CO could induce the diffusion of Pt to the edge of octahedral nanocrystal[39]. In our case, CO may be produced by degrading formaldehyde in the presence of Pt-based nanoparticles. Thus, the Pt atoms may also have tendency to diffuse to the edge of the preformed nano-multipod, followed by the reconstruction via the diffusion of Ni atoms to edge area to form excavated Pt–Ni alloy branches.

The growth of the excavated hexagonal prism can be described by the proposed scheme shown in Fig. 4. First, $Pt(acac)_2$ and $Ni(acac)_2$ are co-reduced into Pt-rich fcc Pt–Ni alloy NCs taking the shape of octahedron. Then, Ni-rich Pt–Ni alloy branches adopting the hcp structure grow along each <111> direction of fcc octahedral cores, and evolve into Pt-Ni multipods consisted of convex hexagonal prisms, accompanying with quick consumption of Ni precursors in the reaction solution. At the final stage, relatively rich Pt precursors in the reaction solution is preferentially deposited on the edge sites of solid branch of hcp Pt–Ni alloy, followed by morphology evolution from solid hexagonal prism to excavated hexagonal prism.

**HER reaction measurement.** The as-prepared hcp Pt–Ni excavated nano-multipods possess a new crystal phase, low Pt content and excavated polyhedral shape with high surface area, which should be a promising low-cost catalyst for large-scale

applications. We then examined the HER performance of the as-prepared products, which is of great significance for water-alkali and chlor-alkali industry but with high energy consuming aroused from sluggish kinetics and high overpotential in basic environment[40–44]. For comparison, commercial Pt/C and fcc Pt–Ni alloy NCs transformed from the hcp Pt–Ni excavated nano-multipods (with the composition and morphology almost unchanged) were chosen as reference catalysts (Supplementary Figs 6 and 19). Figure 5a,b shows the polarization curves of the three catalysts in 0.1 M KOH obtained by linear sweep voltammetry, which display the normalized HER current densities with respect to electrochemically active surface area (ECSA) and the mass of Pt, respectively. At a current density of $10\,mA\,cm^{-2}$, the overpotential for the unique hcp Pt–Ni excavated nano-multipods is only 65 mV versus RHE, much smaller than that of the fcc counterpart and commercial Pt/C. At an overpotential of 70 mV versus reversible hydrogen electrode (RHE), the current densities for the commercial Pt/C, fcc Pt–Ni alloy NCs and as-prepared hcp Pt–Ni excavated nano-multipods are $1.76\,mA\,cm^{-2}$, $5.74\,mA\,cm^{-2}$ and $11.41\,mA\,cm^{-2}$, respectively. It can be seen that the current density of hcp Pt–Ni excavated multipods is 6.5 and 2.0 times higher than those of commercial Pt/C and the fcc counterpart, respectively.

To investigate the catalytic mechanism of the HER, Tafel slope and exchange current density were acquired by fitting the experiment data with the Butler–Volmer equation (Fig. 5c). The Tafel slope of the hcp Pt–Ni excavated multipods and fcc counterpart are $78\,mV\,dec^{-1}$ and $74\,mV\,dec^{-1}$, respectively, which are less than that of the commercial Pt/C ($117\,mV\,dec^{-1}$). The Tafel slope of the commercial Pt/C is comparable to the reported value[45]. For HER in alkaline solution, the dissociation of water (the Volmer reaction) is the rate determining step[41–44]. The lower Tafel slope of Pt–Ni alloy indicates that the alloying of Ni to Pt facilitates HER in alkaline solution. Previously, Markovic and co-workers found $Pt_{0.1}Ru_{0.9}$ exhibits very high activity for HOR/HER in the alkaline environment due to more oxophilic sites on Ru atoms, which facilitate the adsorption of hydroxyl species $(OH_{ad})$[46]. Similar mechanism may be operational in our case, that is, the existence of surface Ni atoms stabilizes the hydroxyl species, thus facilitating the dissociation of water.

Meanwhile, the similarity of the Tafel slope between the hcp Pt–Ni alloy and the fcc counterpart indicate that the two catalysts undergo similar reaction pathways. However, the exchange current density of the hcp Pt–Ni alloy ($1.65\,mA\,cm^{-2}$) is much larger than that of the fcc counterpart and commercial Pt/C ($0.68\,mA\,cm^{-2}$ and $0.46\,mA\,cm^{-2}$, respectively). In addition,

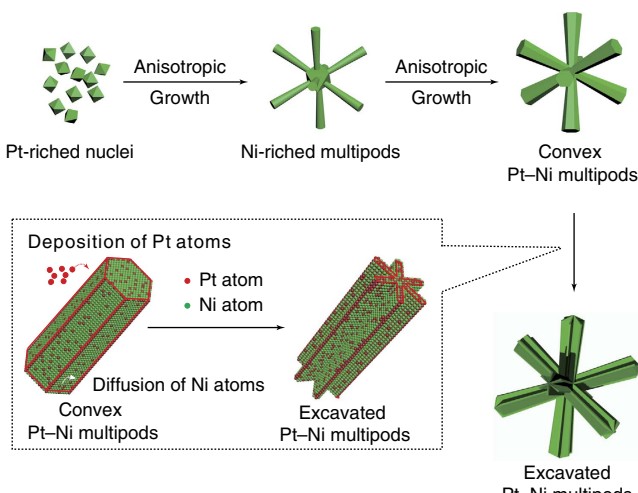

**Figure 4 | The schematic illustration for the formation of Pt–Ni excavated nano-multipods.** Four sections along the direction of arrows (from the top-left to the bottom-right) illustrate the corresponding sequent growth stages, and schematic illustration in the dotted box (bottom-left) represents the morphology evolution from convex hexagonal prism to excavated hexagonal prism.

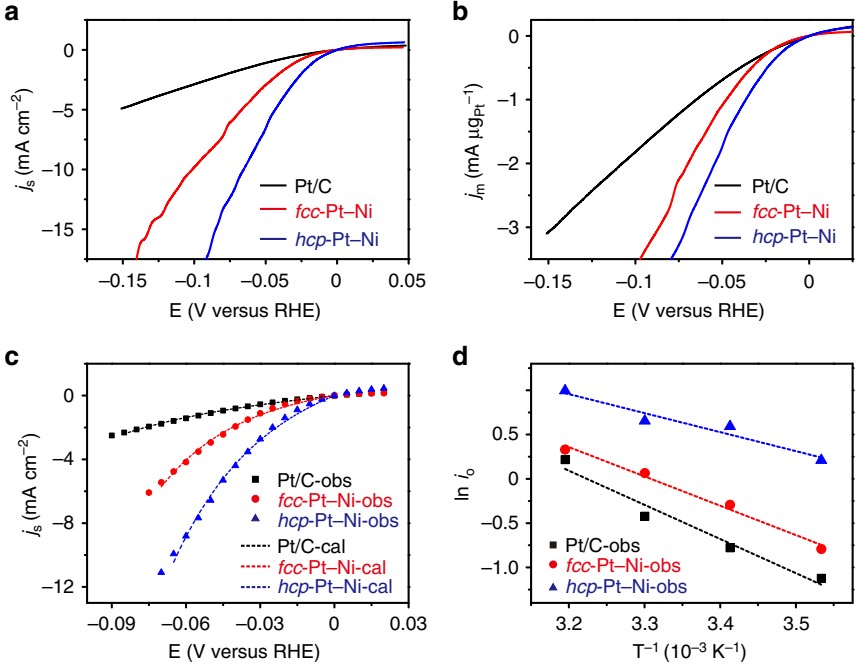

**Figure 5 | Electrochemical measurement of the hcp Pt–Ni excavated nano-multipods.** The HER polarization curves of three catalysts in 0.1 M KOH normalized by (**a**) ECSA and (**b**) mass of Pt. Scanning rate: $10\,mV\,s^{-1}$; rotation rate: 1,600 r.p.m.; temperature: 293 K. (**c**) Measured HER kinetic current densities (scatter dots) and their fittings based on the Butler–Volmer equation (dotted line). (**d**) Representative Arrhenius plots of the HER exchange current densities.

the activation energy of the hcp Pt–Ni excavated multipods was measured to be $17.8 \, kJ \, mol^{-1}$, which was also smaller than that of fcc counterpart ($27.5 \, kJ \, mol^{-1}$) and the commercial Pt/C ($32.0 \, kJ \, mol^{-1}$). The results indicate that the intrinsic activity of the hcp Pt–Ni excavated multipods for HER is better than those of the fcc counterpart and the commercial Pt/C in a basic medium. The enhanced HER activity of the hcp Pt–Ni excavated nano-multipods can be due to the allomorph effect of the Pt–Ni alloy.

Because of the unique hcp phase and low percentage of Pt, the mass current density (at $-70 \, mV$ versus RHE) of the hcp Pt–Ni excavated multipods is $3.03 \, mA \, \mu g_{pt}^{-1}$, which is 2.7 and 1.5 times higher than that of Pt/C and fcc Pt–Ni alloy, respectively. Note that the mass current density of the hcp Pt–Ni excavated multipods outperforms the currently reported catalysts (Supplementary Table 4). To investigate the catalytic durability, $i$–$t$ curves were measured (Supplementary Fig. 21). Although the current density of hcp Pt–Ni excavated nano-multipods dropped a little quickly at beginning of the experiment, the overall and steady current density were much higher than those of the fcc counterpart and the commercial Pt/C. Thus, the hcp Pt–Ni excavated nano-multipods exhibited the best HER performance among the three catalysts, which can be attributed to combination of allomorphism effect, alloy effect and large surface area of the excavated morphology.

## Discussion

We have successfully synthesized excavated Ni-rich Pt–Ni alloy nano-multipods in the hcp phase. The building blocks of the multipods are highly concaved hexagonal prisms assembled by six nanosheets of about 2.5 nm in thickness and with exposed high energy $\{11\bar{2}0\}$ facets. Control studies showed that formaldehyde in the growth solution plays the pivotal role in formation of the unique hcp structure. In addition, the formation of excavated polyhedral morphology can be attributed to the preferential deposition of Pt atom on the edge of the solid branch of hcp Pt–Ni alloy, followed by the diffusion of face-sited Ni atoms to the edges. Benefiting from the unique crystal structure and excavated polyhedral morphology, the highly branched Pt–Ni multipods exhibited a superior catalytic HER activity compared to their fcc counterpart and commercial Pt/C. We believe that syntheses of Pt-based alloy nanomaterials with unusual crystal phases and excavated polyhedral morphologies is a promising approach for developing noble metal-based nanomaterials with enhanced or new functionalities.

## Methods

**Chemicals and materials.** Platinum (II) 2,4-pentanedionate (Pt(acac)$_2$), nickel (II) 2,4-pentanedionate (Ni(acac)$_2$), oleic acid (C$_{18}$H$_{34}$O$_2$, tech. 90%) and Pt/C (20 wt%) were purchased from Alfa Aesar. Formaldehyde solution (40%), and $n$-butylamine were purchased from Sinopharm Chemical Reagent Co. Ltd. (Shanghai, China). Oleylamine (C$_{18}$H$_{37}$N) was purchased from J&K Chemicals. All chemicals were analytical grade and used without further purification.

**Synthesis of hcp Pt–Ni alloy.** In a typical synthesis of Pt–Ni alloy NCs, 8.0 mg of Pt(acac)$_2$ and 15.7 mg of Ni(acac)$_2$ were dissolved in 9.00 ml of oleylamine and 1.00 ml of oleic acid under ultrasonic stirring. An aliquot of 800 μl of formaldehyde solution was injected successively into the mixture stirring after the mixture became a clear solution, then it was further stirred for 15 min and transferred into a Teflon-lined stainless-steel autoclave with a capacity of 20 ml. The sealed vessel was heated from room temperature to 170 °C in about 70 min and kept at this temperature for 12 h, then cooled to room temperature naturally. The products were collected by centrifugation (10,000 r.p.m. for 5 min) and then washed several times with hexane and ethanol to remove impurities.

**Structure and composition characterizations.** The crystal phase of the as-prepared products was determined by powder XRD using a Rigaku Ultima IV X-ray diffractometer with Cu Kα radiation. The Rietveld refinement was conducted by using Topas software. The morphology was observed by the SEM (Hitachi S4800). All samples for the TEM analysis were prepared by depositing a drop of the diluted suspension in ethanol on carbon film-coated copper grid. The HAADF-STEM and EDS were performed with a FEI TECNAI F30 microscope operated at 300 kV. The precise content of every element in sample was determined by the ICP-AES(Baird PS-4).

**Electrochemical measurement.** The as-prepared hcp Pt–Ni alloy excavated nano-multipods (2 mg) were dispersed in 10 ml $n$-butylamine, and 3 mg support materials (Vulcan XC-72 carbon (Premetek Co.)) were dispersed into 5 ml $n$-butylamine to form well-dispersed suspensions. Then the two suspensions were mixed together and stirred for 3 days. The carbon supported hcp Pt–Ni alloy excavated multipods were collected through centrifugation (11,000 r.p.m. for 8 min) and then washed several times with ethanol to remove impurities and dried in the vacuum drying oven (150 °C). To acquire the carbon supported fcc Pt–Ni alloy multipods, post-treatment at 350 °C in the protected atmosphere (5% H$_2$ and 95% N$_2$) were conducted for 12 h.

The carbon supported samples (3.0 mg) or commercial Pt/C (3.0 mg) were dispersed in 1.5 ml ethanol, respectively, and then 15 μl nafion (5%) was added in the ethanol by ultrasound for 0.5 h. All the concentration of catalysts was $2.0 \, mg \, ml^{-1}$. To prepare the catalyst-supported electrode, the glassy carbon electrode (diameter = 5 mm, Pine Instrument) was first polished and washed carefully, and then the as-prepared catalyst suspension (6 μl) was deposited on the pre-treated glassy carbon electrode (the actual amount of Pt on the electrode is 1.5 μg through ICP-AES), which was used as the working electrode after the solvent was vapourized at room temperature.

All electrochemical measurements were recorded using an electrochemical workstation (CHI 660E, Shanghai Chenhua Co., China). In a typical experiment, a Pt slice and Hg/HgO (1 M KOH) were served as the counter electrode and the reference electrode, respectively. Before electrocatalytic experiments were performed, the electrolyte was bubbled by N$_2$ gas for 5 min to achieve the O$_2$-free solution, and then the rotation disk electrode loaded with the catalysts was electrochemically cleaned by continuous potential cycling between $-0.90$ and $-0.4 \, V$ (versus Hg/HgO) at $100 \, mV \, s^{-1}$ in a solution containing 0.1 M KOH (293 K). After that, H$_2$ gas was purged through the solution for 5 min to make the solution saturated with H$_2$. Subsequently, the catalytic activity was measured by linear sweep voltammetry method with a scan rate of $10 \, mV \, s^{-1}$ (at 293 K) while H$_2$ bubbling was continued during the HER activity measurement. The rotation rate of rotation disk electrode was 1,600 r.p.m. to remove the H$_2$ bubble and all the polarization curves were $iR$ corrected.

The ECSAs of the catalysts were determined by the area of the hydrogen desorption peaks in the cyclic voltammetry measurement performed in 0.1 M HClO$_4$ electrolyte with a scan rate of $100 \, mV \, s^{-1}$ (at 293 K). The ECSAs of the catalysts were calculated by the equation ECSA = $Q/q_0$, in which $Q$ is the electric quantity calculated from hydrogen desorption peaks, and $q_0$ is $210 \, \mu C \, cm^{-2}$.

**DFT calculations.** Spin-polarized calculations were carried out by using Perdew–Burke–Ernzerhof[47] gradient-corrected exchange-corrected functional with the projector augmented plane wave[48,49] method as implemented in the Vienna ab-initio simulation package[50,51]. The plane wave kinetic energy cutoff was set to 400 eV. The computational models were described in the Supplementary. For all of the calculations, the vacuum regions between slabs were more than 10 Å, and Monkhurst–Pack $k$-point sampling with $\sim 0.05 \times 2\pi \, Å^{-1}$ spacing in a reciprocal lattice spacing was utilized.

**Data availability.** The data that support the findings of this study are available from the corresponding author on reasonable request.

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

## Acknowledgements

This work was supported by the National Basic Research Program of China (Grant 2015CB932301), the National Natural Science Foundation of China (Grants 21333008, 21603178 and J1030415) and the Natural Science Foundation of Fujian Province of China (No. 2014J01058). We thank Prof Tim Lian (Emory University, USA) for the improvement of English language in the manuscript.

## Author contributions

Z.X. and Y.J. proposed the research project and guided the whole experiments. L.Z. gave important advice on the research project. Z.C. conducted the syntheses and characterization of the hcp Pt–Ni alloy, and drafted the manuscript. Q.C. and W.Z. assisted draft the manuscript and analyse the data. G.F. conducted the theoretical calculations and analyses. S.S. carried out the Rietveld refinement of XRD data. B.L. gave advice on the electrochemical measurement of the hcp Pt–Ni alloy. H.L. helped the characterization of the hcp Pt–Ni alloy.

## Additional information

**Competing interests:** The authors declare no competing financial interests.

**Publisher's note**: 

