## [Peer Review File · Nature Communications]

Reviewers' comments:

Reviewer #1 (Remarks to the Author):

In this manuscript, Z. Cao et al. reported Pt-Ni alloy excavated nano-multipods with hcp crystal structure, and further demonstrated that this materials showed high mass activity towards HER in basic solution. Though these findings are somewhat interesting, this work is not suitable for publishing in Nature Communications due to lack of its academic rigour.

The authors claimed the discovery of hcp Pt-Ni alloy for the first time; however, this conclusion was not supported by the experimental evidences. First of all, the Pt atom percentage in this alloy was quiet low, which was only about 11-12 at%. Furthermore, the initial cores of these multipods were fcc Pt-Ni nanocrystals with a high Pt atom ratio of about 50 at%. These fcc Pt-Ni cores still existed in the final products, proved by its XRD data with an un conspicuous peak located at around 48 degree (Fig 1b). This phenomenon revealed that the Pt percentage might be even lower than 11 at.% in these multipod products. It is well known that nickel is much easily to form its hcp crystal structure (e.g., Nanotechnology, 2006, 17, 3750; Materials Chemistry and Physics, 2005, 89, 359; Advanced Materials, 2013, 25, 1552) Therefore, I strongly suppose that these nano-multipods should be a hybrid of fcc Pt-Ni nanocrystal cores and Pt-doped hcp nickel nanowires.

Therefore, the high mass HER activity of the products in basic solution cannot be arbitrarily attributed to its hcp crystal structure. The manuscript is built on a wrong structural analysis.

Reviewer #2 (Remarks to the Author):

The authors describe the synthesis of a new isomorph of PtNi, i.e. PtNi hcp in the form of highly structural directional nanoparticles. The morphology of the nanoparticles have been studied and related to the proposed formation mechanism. The reported activity is improved over commercial carbon supported Pt nanoparticles both on a surface area normalized basis and on a mass loading basis.

The reported activity and the unique correlation of improved activity to the structure of Pt Ni alloy is of high importance to the field of electrocatalysis and indeed the wider renewable energy community.

It is highly recommended that the authors make/obtain DFT calculations of the altered H-binding energy in the hcp structure as this would greatly add to the impact and the understanding of why this new isomorph is a superior catalyst.

Furthermore, calculations could also support the preferential deposition of Pt onto the edges of Ni-rich octahedral.

The communication currently lacks a thorough explanation of how the multipods are formed. This could be achieved by referencing to the literature for similar systems. For instance, comparing to some of the pioneers in structure directing agents for nanostructures, Alivisatos and Milliron.

Of greatest interest is a more complete understanding of the system being reported. Specifically, what role does each surfactant/reactant play during the hydrothermal synthesis, substantiated with examples from the literature or the authors' experiments.

Could the authors also comment on which alternatives to the Pt-edge deposition hypothesis currently suggested they have investigated. For instance, is it possible that the formaldehyde that is placed in the hydrothermal annealing solution is slowly degrading over time causing the time dependent structure reconstruction. As a results of a change in the growth mechanism?

Finally, some calculations may be helpful in determining which crystal facets have the highest surface energies. This may give some indication of preferential binding or chemistry during synthesis.

Specific comments

In addition to the comparison of activity at a selected overpotential the authors should compare the effect on Tafel slope, and exchange current density. This would allow to discern if the isomorph accesses another rate determining step (Tafel slope) and/or if the intrinsic activity is changed (i.e. exchange current).

Determining the activation energy for the HER would be a great way to unequivocally determine the sources of increased activity and compare this to literature (e.g. DOI 10.1149/1.3483106)

The authors should provide a Rietveld fitting of a PtNi hcp phase and show the fitting parameters to support the assignment.

SEM-EDS mapping should be described in the main text not just shown in Fig.1, in order to show a homogeneous Ni distribution rather than just discussing the overall Ni/Pt ratio.

Reaction mechanism, the authors should use chemical intuition to propose a mechanism explaining the effect of formaldehyde on the reduction.

L. 174. Could the authors elaborate on the evidence that the Pt initially deposits primarily on the edges where it then initiates Ni diffusion. In general, deposition is a competitive process, where different atoms have more or less favorable deposition rates on certain surfaces. Why would Pt prefer deposition on PtNi edges/steps?

L. 217 The authors present the idea that the Pt-Ni hcp catalyst presents the best reported activity for HER in alkaline solution, while the mass activity and ECSA determined activity is certainly improved over Pt/C, there is no comparison to non-Pt based systems. The authors should therefore compare the geometric activity to the state of the arts reports in literature for non-Pt based catalysts such as NiMo (e.g. DOI 10.1021/cs300691m), and Ni5P4(DOI 10.1039/C4EE02940B)

L 283, it is not clear that H₂ bubbling was continued during the HER activity measurements based on the description. As the hydrogen partial pressure is of outmost importance for analysis this close to the reversible potential this has to be maintained to ensure meaningful measurements "After that, H₂ gas was purged through the solution for 5 min. Subsequently, the catalytic activity was measured by linear sweep voltammetry method in a solution containing 0.1 M KOH with a scan rate of 10 mVs⁻¹ (25 oC)."

Fig 1 caption, what pattern is being referenced in b? State or show.

Fig 2 please show the end on view of the excavated multipods, as the description is not obvious.

Fig 3d. Why is it necessary to show the data from 55-90 2theta? Are the weak peaks at 60, 73, and 80 meaningful? They should assigned or marked as unidentified.

Fig 5. What is the dashed vertical line representing? Typically, 10mA/cm² (or 100mV overpotential) is the standard reference point in the community, please clarify the choice of 70mV overpotential.

Minor comments

L 27:" due to totally different atomic arrangements a" it is not clear why the authors would describe the change from one closed-packed-arrangement to another as "totally different". Do the authors mean that a variation in crystal structure may imbue novel properties in the known noble metal catalysts?

L. 58 "alloy, which can be" should be "alloy, but can be "

L. 197 Please clarify if the reported values are on a geometric, ECSA normalized, or mass

basis.

L. 262 typo "butlamine" should probably be butylamine.

Reviewer #3 (Remarks to the Author):

This work shows the unique synthesis of Pt-Ni nano-multipod with an uncommon hcp phase and its superior activity towards hydrogen evolution reaction in alkaline solution. The authors carefully explored the evolution mechanism of the excavated multipod structure from the Pt-Ni octahedral nanocrystals, which provides insights into the synthesis and design of nanocatalysts with specific structures and morphologies. This is a decent research work and may be accepted for publication once the following issues are addressed.

1. In the synthesis of excavated Pt-Ni nano-multipods, the Ni deposition dominates from 3 h to 9 h of the process and the composition of Pt keeps dropping in the meantime (from 50 % to 9 %). The hcp phase formed during this period seems to mainly come from the Ni component. Is it because Ni helps build the hcp crystal phase? Is the 9 h product rich of Ni at the surface?
2. Considering that hcp phase is metastable, was it still maintained well after HER test? And could you add the TEM image of the transformed fcc Pt-Ni alloy from the hcp Pt-Ni excavated nano-multipods upon heat-treatment?
3. In Figure S12, we can see that the current density of the hcp Pt-Ni excavated nano-multipods dropped from $\sim 10 \text{ mA cm}^{-2}$ to $\sim 5 \text{ mA cm}^{-2}$ in the first 3000 s, though they were claimed to be stable. However, the HER performance of Pt/C and fcc Pt-Ni alloy looks relatively stable.

Reviewer 1

Comments: In this manuscript, Z. Cao et al. reported Pt-Ni alloy excavated nano-multipods with hcp crystal structure, and further demonstrated that this materials showed high mass activity towards HER in basic solution. Though these findings are somewhat interesting, this work is not suitable for publishing in Nature Communications due to lack of its academic rigour.

The authors claimed the discovery of hcp Pt-Ni alloy for the first time; however, this conclusion was not supported by the experimental evidences. First of all, the Pt atom percentage in this alloy was quite low, which was only about 11-12 at%. The initial cores of these multipods were fcc Pt-Ni nanocrystals with a high Pt atom ratio of about 50 at%. These fcc Pt-Ni cores still existed in the final products, proved by its XRD data with an un conspicuous peak located at around 48 degree (Fig 1b). This phenomenon revealed that the Pt percentage might be even lower than 11 at.% in these multipod products. It is well known that nickel is much easily to form its hcp crystal structure (e.g., Nanotechnology, 2006, 17, 3750; Materials Chemistry and Physics, 2005, 89, 359; Advanced Materials, 2013, 25, 1552). Therefore, I strongly suppose that these nano-multipods should be a hybrid of fcc Pt-Ni nanocrystal cores and Pt-doped hcp nickel nanowires. The high mass HER activity of the products in basic solution cannot be arbitrarily attributed to its hcp crystal structure. The manuscript is built on a wrong structural analysis.

Response: Thanks very much for the reviewer's comment. We think the revised manuscript could meet the academic criterion of Nature Communications. The detailed responses are listed in the following:

About the *hcp* phase of Pt-Ni alloy:

We do not agree that our products are Pt-doped *hcp* nickel nanowires.

“Doping” is usually used for describing the materials with very low concentration of dopant. For example, the concentration of dopant is usually in ppb level for semiconductor doping. Alloy is the product of two and more metal, which is divided into three major categories: metallic solid solution, metallic compounds, and metallic interstitial compounds. To the best of our knowledge, the composition of solid solution can vary at a large range. For example, the content of carbon atoms in the ferrite (iron alloy) is about 0.1 at%, while the copper in Au-Cu alloy could vary from *ca.* 0 at% to *ca.* 100 at%.

We agree that *hcp* phase of Pt-Ni alloy has relationship with *hcp* nickel phase. From the phase diagram point of view, an alloy phase could be the same as a pure metal phase. Therefore, it is normal that the Pt-Ni alloy adopts the same phase of pure *hcp* Ni phase. From this study, we can only obtain the *hcp* phase Pt-Ni alloy when the Pt content is low (e.g. 12 at%), which indicate that the *hcp* Pt-Ni alloy phase has some relationship with *hcp* nickel phase. However, it is not Pt-doped nickel.

About the content of *hcp* Pt-Ni alloy phase:

Based on the XRD data, we conducted the *Rietveld* refinement, which is very useful for quantitative analysis (which has been added in the revised manuscript, L35-L39, page 2). The results showed that the final product was composed of 99.8 wt% of *hcp* Pt-Ni alloy phase and 0.2 wt% *fcc* Pt-Ni alloy phase (please see Table S1 in the supporting information of revised manuscript). The content of *fcc* PtNi alloy phase is very low in the products, which would not affect the properties of *hcp* Pt-Ni multipods. In addition, we carefully detected the content of Pt in the single *hcp* Pt-Ni alloy branch by using EDS technique, we find it is same as overall content (as shown in Figure S3 in the supporting information). Therefore, it is reasonable to define the final product Pt-Ni multipods as an *hcp* Pt-Ni alloy.

About attributing the high mass HER activity of the products to its *hcp* crystal structure:

As discussed above, the content of *hcp* Pt-Ni alloy is 99.8 wt%. The *fcc* Pt-Ni alloy core is only 0.2 wt%, and was surrounded by several branches of *hcp* Pt-Ni phase, and thus the *fcc* phase in the product should be negligible in the hydrogen evolution reaction. In addition, we chose the *fcc* Pt-Ni alloy counterpart as a reference, which was transformed from the as-prepared *hcp* Pt-Ni catalyst with the morphology and composition unchanged. Since the two catalysts possess a similar morphology and the same composition, it is reasonable to attribute the high HER activity of the products to its *hcp* crystal structure.

Reviewer 2

The authors describe the synthesis of a new isomorph of PtNi, i.e. PtNi hcp in the form of highly structural directional nanoparticles. The morphology of the nanoparticles have been studied and related to the proposed formation mechanism. The reported activity is improved over commercial carbon supported Pt nanoparticles both on a surface area normalized basis and on a mass loading basis. The reported activity and the unique correlation of improved activity to the structure of PtNi alloy is of high importance to the field of electrocatalysis and indeed the wider renewable energy community.

Response: We kindly thank the reviewer for the positive feedback and constructive suggestion. According to these comments, we have carefully revised our manuscript, and the detailed revisions are highlighted in the text.

1. It is highly recommended that the authors make/obtain DFT calculations of the altered H-binding energy in the hcp structure as this would greatly add to the impact

and the understanding of why this new isomorph is a superior catalyst. Furthermore, calculations could also support the preferential deposition of Pt onto the edges of Ni-rich octahedral.

Response: Thanks a lot for the valuable suggestion. For HER in the alkaline solution, the rate-determining step for the Pt-group catalyst is the *Volmer* reaction ($\text{H}_2\text{O} + \text{e}^- \rightarrow \text{H}_{\text{ads}} + \text{OH}^-$) because the H-binding energy (determining the *Tafel* reaction or *Heyrovsky* reaction) for Pt-group materials is nearly optimal (*Science* 2011, **334**, 1256; *Nat Mater.* 2012, **11**, 550; *Angew. Chem. Int. Ed.* 2012, **51**, 12495). Following the reviewer's suggestion, we also carried out the DFT calculation for the H-binding energy on different site of both *hcp* and *fcc* Pt-Ni alloy (as shown in following Figure and Table), and found the difference of H-binding energy between the different crystal structure (*hcp* and *fcc*) is very small.

Figure C1. Different surface sites on: (a) *fcc* {110}; (b) *hcp* {11 $\bar{2}$ 0}.

Table C1. Summary of the calculated adsorption energies for H on different surfaces

crystal phase	adsorption site	adsorption energy of H atom / eV		
		Ni(110)	Ni ₅₀ Pt ₅₀ (110)	Pt skin
fcc	Top	-2.19	-2.19/-2.64	-2.13
	Short bridge	-2.65	-2.71	-2.70
	Long bridge	-2.68	-2.49	-2.70
	Hollow	-2.72	-2.71	-
hcp	Top	-2.25	-2.64	-2.71
	Short bridge	-2.78	-2.70	-2.71
	Long bridge	-2.75	-2.63	-2.70
	Hollow	-2.79	-2.70	-

Considering the *Volmer* reaction on the Pt-Ni surface, we had tried to carry out DFT calculation to understand the deep mechanism. However, it is very difficult to construct a surface model as there are many possible local interfacial structures for the Pt₁₂Ni₈₈ alloy. Currently, the exact surface structure of Pt-Ni alloy is difficult to obtain, and thus we are not able to carry out the calculation.

As for the preferential deposition of Pt onto the edge sites, it was affected by the chemical environment. The calculation would also be difficult due to the difficulty in building calculation model.

2. The communication currently lacks a thorough explanation of how the multipods are formed. This could be achieved by referencing to the literature for similar systems. For instance, comparing to some of the pioneers in structure directing agents for nanostructures, Alivisatos and Milliron.

Response: Following the reviewer's suggestion, we surveyed the references similar to our synthesis system. A. P. Alivisatos and the co-workers proposed that the polymorphism existing in different part of the nanostructure combined with the regulation of growth kinetics can facilitate the formation of branched nanostructures (DOI: 10.1038/nmat902). Our case could be similar to the above mentioned growth mechanism. Firstly, the *fcc* octahedral Pt-rich nano-particles formed as cores of the branched nanostructure (Figure 3a and 3d). As the atomic arrangement of {111} facets of *fcc* phase is identical to {001} facets of *hcp* lattice, the Ni-rich Pt-Ni alloy (*hcp* phase) then grows epitaxially on the {111} facets with the low energy barrier. In addition, we found the branched structure can only be found when the *hcp* appeared (Figure S11a and S12). Thus, anisotropic growth of Pt-Ni alloy nanocrystal into nanobranch could be due to the regulation of growth kinetics because of anisotropic property along c-axis of the *hcp* structure. In addition, selective adsorption of oleic acid on specific crystal facets may also affect the growth kinetics, as the branches cannot be formed in the absence of oleic acid (Figure S14). Relevant discussion was added in the revised text (Lines 42-44, page 5 and Lines 1-7, page 6).

3. Of greatest interest is a more complete understanding of the system being reported. Specifically, what role does each surfactant/reactant play during the hydrothermal synthesis, substantiated with examples from the literature or the authors' experiments.

Response: Thanks a lot for the reviewer's comments. In our synthesis, the oleylamine served as the solvent, and the formaldehyde solution played the role of reductant and the structure inducing agent. When the amount of the formaldehyde solution was less than 200 μ L, the product was *fcc* Pt-Ni alloy. However, when the amount of formaldehyde solution increased to more than 400 μ L, the product turned to be Pt-Ni *hcp* alloy. Therefore, formaldehyde played a key role in the formation of *hcp* Pt-Ni phase. Besides, we investigated the role of oleic acid and found it is important in the formation of branched structures. When no oleic acid was added, the final product

was *hcp* Pt-Ni irregular nanoparticle (Figure S14 and S15) rather than the unique nano-multipods. The result suggested oleic acid was likely to act as a surface stabilization agent by selectively adsorbing to side crystal facet of the hexagonal prism. The added discussion was highlighted in lines 5-7, page 6 following Figure S15 in the Supporting Information.

4. Could the authors also comment on which alternatives to the Pt-edge deposition hypothesis currently suggested they have investigated. For instance, is it possible that the formaldehyde that is placed in the hydrothermal annealing solution is slowly degrading over time causing the time dependent structure reconstruction. As a results of a change in the growth mechanism?

Response: Thanks for the suggesting comment. For the tendency of Pt to deposit on the edges of PtNi crystallite, many experimental facts show that edges of polyhedral nanocrystallites are rich of Pt in the Pt-Ni alloy nanocrystals. For example, in the growth of octahedral Pt-Ni alloy, the edges of the octahedral nanoparticles were Pt-rich (DOI: 10.1126/science.1261212), which was directly observed by EDS mapping. Similar growth mechanism was also reported in the formation of rhombic dodecahedron Pt-Ni alloy, where the Pt always deposit or migrate to the edge of the rhombic dodecahedron leading to the formation of Pt-rich frame enclosing a Ni rich interior phase (DOI: 10.1038/nmat4724). This is also confirmed the evolution process of Pt₃Ni nanoframe (DOI: 10.1126/science.1249061). In our case, EDS mapping of the product formed with 9 hours reaction also shows similar result (Figure S17). It should be noted that no detailed mechanism was proposed to explain this preferential deposition to date, and it needs further investigation.

On the other hand, there was report that the presence of CO could induce the diffusion of Pt to the edge of octahedral (DOI:10.1021/nn5068539). In our case, CO may be produced by degrading formaldehyde in the present of noble metal particles. Thus, the Pt atom may also have tendency to diffuse to the edge of the preformed nano-multipod.

The above discussion and related references has been added in the revised manuscript (Lines 15-23, page 7).

5. Some calculations may be helpful in determining which crystal facets have the highest surface energies. This may give some indication of preferential binding or chemistry during synthesis.

Response: Thanks for the valuable comment. The surface energy of basic crystal facet can be evaluated by the coordination number of surface atoms. For the *hcp* crystal structure, the coordination numbers of (001), (100) and (110) facets are 9, 8 and 7, respectively. Therefore, the (110) crystal facets in an *hcp* structure have very high surface energy due to its low coordination number of surface atom. In the revised

manuscript, we briefly described the sequence of surface energy of these basic crystal facets (Lines 14-17, page 3) and the corresponding models were shown in Figure S4.

Specific comments and corresponding response.

1. In addition to the comparison of activity at a selected overpotential the authors should compare the effect on Tafel slope, and exchange current density. This would allow to discern if the isomorph accesses another rate determining step (Tafel slope) and/or if the intrinsic activity is changed (i.e. exchange current).

Response: Thanks a lot for the constructive comments. The *Tafel* slopes and exchange current densities were calculated by fitting the experimental data to the *Butler-Volmer* equation in the revised manuscript (Figure 5c). The detailed discussion can be found in lines 27-38 of page 8.

2. Determining the activation energy for the HER would be a great way to unequivocally determine the sources of increased activity and compare this to literature (e.g. DOI 10.1149/1.3483106)

Response: Thanks a lot for the suggestion. Following to the method reported in the literature (e.g. DOI: 10.1149/1.3483106), we determined the activation energy of the *hcp* Pt-Ni excavated multipods ($17.8 \text{ kJ}\cdot\text{mol}^{-1}$) and *fcc* counterpart ($27.5 \text{ kJ}\cdot\text{mol}^{-1}$). The value for the commercial Pt/C was $32.0 \text{ kJ}\cdot\text{mol}^{-1}$, which coincide with the reported value in the literature. The description and the discussion can be found in Figure 5d and lines 38-43 of page 8 and lines 1-2 of page 9 in the revised text.

3. The authors should provide a Rietveld fitting of a PtNi hcp phase and show the fitting parameters to support the assignment.

Response: Thanks a lot for the suggestion. Following the reviewer's suggestion, we first re-measured the XRD pattern with a very low scan rate in order to get high quality data. Then, the *Rietveld* refinement (Topas software) was conducted by applying the *hcp* Pt-Ni alloy structure model. The reliability factors of R_p and R_{wp} are 2.05 and 2.97, respectively, which indicates our structure model is corrected. The fitting result was shown in Figure 1b, and the refined structure parameters were summarized in Table S1. In addition, due to the fact that a small diffraction peak at 48.6° (corresponding to the *fcc* phase) appeared, we applied multi-phase simulation, in which both *fcc* and *hcp* phases were considered. The refinement demonstrated that the final product was composed of 99.8 wt% *hcp* phase ($\text{Pt}_{12}\text{Ni}_{88}$) and 0.2 wt% *fcc* phase ($\text{Pt}_{56}\text{Ni}_{44}$). It should be noted that the 0.2 wt% of *fcc* Pt-Ni phase was the early formed cores of the nanostructures. The corresponding revisions can be found in lines 35-39 of page 2 in the revised text.

4. EDS mapping should be described in the main text not just shown in Fig.1, in order to show a homogeneous Ni distribution rather than just discussing the overall Ni/Pt ratio.

Response: Thanks a lot for the suggesting comments. We added the corresponding description about the EDS mapping (lines 33-34, page 2).

5. L. 174. Could the authors elaborate on the evidence that the Pt initially deposits primarily on the edges where it then initiates Ni diffusion. In general, deposition is a competitive process, where different atoms have more or less favorable deposition rates on certain surfaces. Why would Pt prefer deposition on PtNi edges/steps?

Response: Thanks for the comments. We do not understand why Pt prefers to deposit on the edges of Pt-Ni crystallite at the current stage. However, a lot of experimental facts show that edges of polyhedral nanocrystallites are rich of Pt in the Pt-Ni alloy nanocrystals. For example, in the growth of octahedral Pt-Ni alloy, the edges of the octahedral nanoparticles were Pt-rich (DOI: 10.1126/science.1261212), which were directly observed by EDS mapping. Similar growth mechanism was also reported in the formation of rhombic dodecahedron Pt-Ni alloy, where the Pt always deposit or migrate to the edges of the rhombic dodecahedron leading to the formation of Pt-rich frame enclosing a Ni-rich interior phase (DOI: 10.1038/nmat4724). This is also confirmed by the evolution process of Pt₃Ni nanoframe (DOI: 10.1126/science.1249061). In our case, EDS mapping of the product formed with 9 hours reaction also shows similar result (Figure S17). It should be noted no detailed mechanism was proposed to explain this preferential deposition to date, and it is also difficult for us to understand the growth process.

There was also report that the presence of CO could induce the diffusion of Pt to the edge of octahedral (DOI:10.1021/nn5068539). In our case, CO may be produced by degrading formaldehyde in the present of noble metal particles. Thus, the Pt atom may also have tendency to diffuse to the edge of the preformed nano-multipod. The detailed discussion has been added in the revised manuscript (Lines 15-23, page 7)

6. L. 217 The authors present the idea that the Pt-Ni hcp catalyst presents the best reported activity for HER in alkaline solution, while the mass activity and ECSA determined activity is certainly improved over Pt/C, there is no comparison to non-Pt based systems. The authors should therefore compare the geometric activity to the state of the arts reports in literature for non-Pt based catalysts such as NiMo (e.g. DOI 10.1021/cs300691m), and Ni₅P₄(DOI 10.1039/C4EE02940B)

Response: Thanks a lot for the reviewer's comments. Following the reviewer's suggestion, we surveyed the relevant literatures including NiMo (DOI: 10.1021/cs300691m) and Ni₅P₄ (DOI: 10.1039/C4EE02940B). For comparison, we

summarized the catalytic performances of different catalysts in Table S2, where the current density was normalized to mass, ECSA, geometric area. It can be found the loading amount of catalyst of non-Pt based systems like NiMo and Ni₅P₄ were about 1000 times (even 50000 times) more than that of the *hcp* Pt-Ni alloy. However, the current density normalized to the geometric area was still lower than that of the unique *hcp* Pt-Ni alloy.

7. L 283, it is not clear that H₂ bubbling was continued during the HER activity measurements based on the description. As the hydrogen partial pressure is of utmost importance for analysis this close to the reversible potential this has to be maintained to ensure meaningful measurements "After that, H₂ gas was purged through the solution for 5 min. Subsequently, the catalytic activity was measured by linear sweep voltammetry method in a solution containing 0.1 M KOH with a scan rate of 10 mVs⁻¹ (25 °C)."

Response: We agree with the reviewer's comments. Keeping the hydrogen partial pressure as a constant in LSV test is of utmost importance for the HER. Therefore, the hydrogen partial pressure was kept unchanged throughout the HER experiment. The detailed experimental description was not clear in the previous manuscript. We have revised the description (Lines 13-16, page 11).

8. Fig 1 caption, what pattern is being referenced in b? State or show.

Response: Thanks a lot for the kind suggestion. The black rod-lines in the original manuscript presented the calculated values of diffraction peaks for the product *hcp* Pt-Ni alloy, not the diffraction peaks from any reference. In the revised manuscript, we conducted the *Rietveld* refitting on the XRD pattern (as shown in revised Figure 1b), the black rod-lines were deleted, and the fitting data was added instead.

9. Fig 2 please show the end on view of the excavated multipods, as the description is not obvious.

Response: Thanks for the comments. Each branch of multipod was excavated and composed of six ultrathin nanosheets as shown in the model of Figure 2. According to the reviewer's suggestion, we added the number to denote every nanosheet in the schematic model and the corresponding parts of TEM images in the revised manuscript. By this way, the description could become clear. The revised description was highlighted in lines 18-21 of page 3 and lines 1-12 of page 4.

10. Fig 3d. Why is it necessary to show the data from 55-90 2theta? Are the weak peaks at 60, 73, and 80 meaningful? They should assigned or marked as unidentified.

Response: Thanks for the constructive suggestion. In the revised manuscript, we show the diffraction angle from 35 to 55 degree for Fig. 3d.

11. Fig 5. What is the dashed vertical line representing? Typically, 10mA/cm² (or 100mV overpotential) is the standard reference point in the community, please clarify the choice of 70mV overpotential.

Response: Thanks for the comment. In the reference 41 (DOI: 10.1038/ncomms7430), the authors summarized many outstanding studies and selected 70 mV as the reference point. In order to better compare our catalysts with those reported data, we thus adopted this reference point. Following the reviewer's suggestion, we also point out the value of overpotential of our catalysts at 10 mA·cm⁻² in the revised manuscript (Lines 20-23, page 1 and Lines 20-22, page 8).

Minor comments

*1. L 27:" due to totally different atomic arrangements a" it is not clear why the authors would describe the change from one closed-packed-arrangement to another as *totally different". Do the authors mean that a variation in crystal structure may imbue novel properties in the known noble metal catalysts?*

Response: Thanks for the comment. We agree that "totally different atomic arrangements" may not be suitable as both *fcc* and *hcp* phases are closed-packed arrangement. Here, in the introduction part, we would like to highlight different phase has different atomic arrangement, which will result in different properties. In the revised manuscript, it has been revised as "due to different atomic arrangements" (Lines 27, page 1), which would be more adequate.

2. L. 58 "alloy, which can be" should be "alloy, but can be "

Response: Thanks. The correction was made and highlighted in lines 16-17 of page 2 in the revised text.

3. L. 197 Please clarify if the reported values are on a geometric, ECSA normalized, or mass basis.

Response: Thanks for the kind suggestion. The current density in Figure 5a and Figure 5b were normalized to the ECSA and the mass of Pt on the electrode, respectively. We have clarified them in the revised manuscript (Lines 18-19 of page 8).

4. L. 262 typo "butlamine" should probably be butylamine.

Response: Thanks. We have corrected the typos (Line 31, page 10).

Reviewer 3

Comments: This work shows the unique synthesis of Pt-Ni nano-multipod with an uncommon hcp phase and its superior activity towards hydrogen evolution reaction in alkaline solution. The authors carefully explored the evolution mechanism of the excavated multipod structure from the Pt-Ni octahedral nanocrystals, which provides insights into the synthesis and design of nanocatalysts with specific structures and morphologies. This is a decent research work and may be accepted for publication once the following issues are addressed.

Response: We kindly thank the reviewer for the positive feedback and constructive suggestion. According to these comments, we have carefully revised our manuscript, and the revisions were highlighted in the text.

1. In the synthesis of excavated Pt-Ni nano-multipods, the Ni deposition dominates from 3 h to 9 h of the process and the composition of Pt keeps dropping in the meantime (from 50 % to 9 %). The hcp phase formed during this period seems to mainly come from the Ni component. Is it because Ni helps build the hcp crystal phase? Is the 9 h product rich of Ni at the surface?

Response: Thanks for the comments. For the product of 3h, there are only octahedral core with fcc Pt-Ni alloy. In this stage, the content of Pt is high (c.a. 50 at%). After 3 h, hcp Pt-Ni alloy branches with low Pt content (c.a. 12 at%) grew on the octahedral core. The decreasing of Pt content with the reaction time is due to the growth of hcp Pt-Ni alloy branches with low Pt content.

In addition, we carried out a control experiment to further demonstrate whether the content of Ni could affect the formation of hcp Pt-Ni phase by changing the molar ratio between precursors Pt(acac)₂ and Ni(acac)₂. It was found the hcp phase Pt-Ni alloy can only be obtained when the Pt content is low, which indicate that the Ni helps build the hcp crystal phase (Figure S11 and S12). The corresponding experimental results and discussion have been added in lines 16-25 of Page 5 in maintext and Figure S11 and S12 of the supporting information.

2. Considering that hcp phase is metastable, was it still maintained well after HER test? And could you add the TEM image of the transformed fcc Pt-Ni alloy from the hcp Pt-Ni excavated nano-multipods upon heat-treatment?

Response: According to the reviewer's comments, we have carried out additional analysis of *hcp* Pt-Ni alloy catalyst after HER test. The TEM, HRTEM images of the XC-72 supported *hcp* Pt-Ni alloy catalyst after HER test were characterized (shown in Figure S21 in the supporting information of revised manuscript). It can be found that the *hcp* structure was well maintained after the catalytic reaction. In addition, the TEM image of the *fcc* Pt-Ni alloy counterpart transformed from the *hcp* Pt-Ni excavated multipods upon heat-treatment has been added as Figure S18 in the supporting information, which shows that the morphology kept unchanged.

3. In Figure S12, we can see that the current density of the hcp Pt-Ni excavated nano-multipods dropped from $\sim 10 \text{ mA cm}^{-2}$ to $\sim 5 \text{ mA cm}^{-2}$ in the first 3000 s, though they were claimed to be stable. However, the HER performance of Pt/C and fcc Pt-Ni alloy looks relatively stable.

Response: Thanks for the comment. The description about the stability was a little inaccurate. Although the current density of *hcp* Pt-Ni excavated nano-multipods drops a little quickly at the beginning, the overall and steady current density was much higher than that of the *fcc* counterpart and the commercial Pt/C. We have corrected the description of the catalytic stability in a more proper way (Lines 10-14 of page 9, and Figure S20).

Reviewers' comments:

Reviewer #1 (Remarks to the Author):

In this revised manuscript, the authors had clearly demonstrated that the Pt species were well-distributed in this Pt-Ni alloy nanostructure with hcp phase by XRD and EDX characterization. This phenomenon indicated the Pt atoms in the initial Pt-riched nuclei should happen to migrate during the following growth process, which was similar to recent findings of P. Yang et al. that anisotropic phase segregation and migration of Pt in the formation of Pt–Ni nanocrystals (Nature Materials, 2016, 15, 1188, also as Ref. 35 in this revised version). Therefore, I suggest the acceptance of this manuscript for publication in Nature Communications. The migration mechanism of Pt species in this Pt-Ni alloy with unusual phase is also highly suggested to investigate in the subsequent research.

Reviewer #2 (Remarks to the Author):

The authors have addressed most of the comments posed by this reviewer, below is a clarification of the residual comments. In addition, based on the comment by Reviewer 1 we have added the below point to assist the authors clarifying the difference between hcp-Pt-Ni to which the reviewer correctly objects and Pt alloyed into hcp Ni which seems to be the case in this work.

R1 C1: L 157 Regarding reviewer #1's comments on the accuracy of terming this catalyst hcp Pt-Ni alloy. As an example, this reviewer finds that statements such as: "The results indicate that the Ni helps build the hcp crystal phase." Are misleading, the compound synthesized appears to be hcp-Ni alloyed with small quantities of Pt, thus Ni does not "help" but rather it tolerates small quantities of Pt substituted in the host Ni-lattice structure, it is not Pt that adopts an hcp structure.

R2 C1: The authors state in their rebuttal: "As for the preferential deposition of Pt onto the edge sites, it was affected by the chemical environment. The calculation would also be difficult due to the difficulty in building calculation model.". The authors are suggesting on L 173-182 that Ni hcp structure preferentially adsorbs on the fcc Ni-Pt cube core on the [111] facet, this should be possible for the authors to determine the relative energy of a [111] slab with Ni adsorbate atoms (to monolayers) relative to that of Pt adsorption.

Finally, the authors present enthalpy of formation for H-binding whereas the common procedure in the HER field is to use the computational H-electrode model to correct the raw calculated quantities and give an improved comparison to literature. The authors should seriously consider using this terminology/methodology as the numbers currently have little meaning to the general audience of Nat Commun.

R2 C3: The authors appears to have misunderstood the comment made, the authors should consider adding literature references/DFT calculations to support the preferred adsorption of oleic acid on the facets that would result in the structure observed, not merely state that it empirically is a required co-solvent(structure directing agent).

R2 C5: The authors bring up an interesting point here. Have the authors considered if the low coordination of the [110] face of hcp-Ni could favor the binding of structural directing agents such as oleic acid.

R2 C6 L. 257 The authors state: "However, the exchange current density of the *hcp* Pt-Ni alloy (1.65 mA·cm⁻²) is much larger than that of the *fcc* counterpart and commercial Pt/C (0.68 mA·cm⁻² and 0.46 mA·cm⁻², respectively)." The authors should compare the exchange current density on the basis of the ECSA not only the geometric surface area. As the j_0 is proportionally dependent on surface area this will allow for a more direct comparison of intrinsic activity. The discussion of the Tafel slope is very useful, the authors should compare the effect of adding Ni to the studies by Markovic (DOI doi:10.1038/nchem.1574) in which alkaline HER is thoroughly discussed.

Reviewer #3 (Remarks to the Author):

The authors answered my questions properly and I have no further concerns on the paper.
The paper is acceptable for publication.

Comments: R1 C1: L 157 Regarding reviewer #1's comments on the accuracy of terming this catalyst hcp Pt-Ni alloy. As an example, this reviewer finds that statements such as: "The results indicate that the Ni helps build the hcp crystal phase." Are misleading, the compound synthesized appears to be hcp-Ni alloyed with small quantities of Pt, thus Ni does not "help" but rather it tolerates small quantities of Pt substituted in the host Ni-lattice structure, it is not Pt that adopts an hcp structure.

Response: Thanks a lot for the comment. In the statement "The results indicate that the Ni helps build the hcp crystal phase", the word "help" may not be appropriate. In fact, we would like to say that the final hcp Pt-Ni alloy has relationship with hcp nickel phase (as pointed out in our previous response to reviewer #1's comments, "We agree that hcp phase of Pt-Ni alloy has relationship with hcp nickel phase.").

For the term "hcp Pt-Ni alloy", we think it is appropriate. Alloy is the product of two and more metals, which is classified into metal solid solution, metallic compounds, and metallic interstitial compounds. The as-prepared product is hcp Pt-Ni metal solid solution based on the phase characterization and the elemental analysis. In a metal solid solution, the content of "solute" might be limited. In our case, the Pt content is 12 at% (31 wt%), and in fact the Pt content in the hcp Pt-Ni alloy could reach 25 at% (52 wt%) according to our experimental result in another synthesis condition. Therefore, the content of Pt is not "small quantity".

On the other hand, from viewpoint of the phase diagram, only a few bi-metal alloys form a single phase; most bi-metal alloys form more than one phase with the variation of their composition. For all the phases in the phase diagram (except the pure single metal phase), we use the term "A-B alloy" (such as Ni-Pt alloy). We usually do not denote them as "A-phase alloyed with B" (such as hcp-Ni alloyed with small quantities of Pt).

According to the reviewer's comments, in the revised manuscript, we revised the corresponding description as: "The results indicate that hcp phase of Pt-Ni alloy has relationship with hcp nickel phase." The revised discussion was highlighted in the text (L26-27, Page 5).

Comments: R2 C1: The authors state in their rebuttal: "As for the preferential deposition of Pt onto the edge sites, it was affected by the chemical environment. The calculation would also be difficult due to the difficulty in building calculation model." The authors are suggesting on L 173-182 that Ni hcp structure preferentially adsorbs on the fcc Ni-Pt cube core on the [111] facet, this should be possible for the authors to determine the relative energy of a [111] slab with Ni adsorbate atoms (to monolayers) relative to that of Pt adsorption.

Response: Thanks a lot for the valuable suggestion. Following the reviewer's suggestion, we carried out the DFT calculation to explore the deposition energies of Ni or Pt atoms on the fcc Pt-Ni {111} facet. By simply considering the adsorption of Ni or Pt atoms on the Pt-Ni {111} surface, the calculation results show that the adsorption energy of Pt atoms are always more favorable, which cannot account our

experiments. We assumed that the reduced metal atoms should be coordinated with solvent molecules. Our calculations showed that binding energy of amine (the solvent) with reduced Pt atoms is much stronger than that with reduced Ni atoms and thus the Pt atoms would have much lower chemical potential than Ni atoms in solvent (the calculation details and explanation are shown in Supplementary Table 3). As a result, a reverse preference of deposition could occur, nicely explaining the formation of Ni-rich Pt-Ni alloy in our experiments. We then expect that Pt atoms could keep preferential reduction and deposition in a solvent with weak coordination ability. Accordingly, a new experiment was designed by replacing the oleylamine with octadecene. Interestingly, it was observed that Pt-rich Pt-Ni alloy formed in the similar condition (Supplementary Fig 16), echoing with theoretical prediction. Relevant discussion has been added in the revised manuscript (L4-11, Page 7 and Supplementary Table 3).

Comments: R2 C1: Finally, the authors present enthalpy of formation for H-binding whereas the common procedure in the HER field is to use the computational H-electrode model to correct the raw calculated quantities and give an improved comparison to literature. The authors should seriously consider using this terminology/methodology as the numbers currently have little meaning to the general audience of Nat Commun.

Response: Thanks a lot for the comment. The computational H-electrode model is very helpful to understand the HER activity in acidic medium. However, here we investigate the HER performance in the basic medium. The rate determining step would involve the dissociation of water, which is very different from that in acidic medium. Thus, we did not discuss them from this aspect in our manuscript.

In addition, we are sorry that we did not very clearly state our thought in the previous response to the reviewers C1. In the previous response, we stated that “HER in the alkaline solution, the rate-determining step for the Pt-group catalyst is the *Volmer* reaction ($\text{H}_2\text{O} + \text{e}^- \rightarrow \text{H}_{\text{ads}} + \text{OH}^-$) because the H-binding energy (determining the *Tafel* reaction or *Heyrovsky* reaction) for Pt-group materials is nearly optimal”. We would like to express that the H-binding energy is not key for the HER in the alkaline solution, and we would not like to discuss from this aspect. However, we have misled the reviewer because we listed some of our DFT calculation about the H-binding energy in the following part of the response to C1, although we did not include them in the manuscript and the supporting information in the previous manuscript.

For the H-electrode model, *Gibbs* free energy (ΔG_{H}) is more suitable to describe the adsorption of hydrogen. Under acidic condition, the optimized ΔG_{H} for HER should be much more close to 0 eV. It can be seen from Table C1 that the ΔG_{H} for the H adsorption energy on the *hcp* $\{11\bar{2}0\}$ is a little bit stronger than that on *fcc* $\{110\}$.

However, in the basic medium, the rate determining step of HER involves the dissociation of water rather than the H atom coupling. In this case, a stronger ΔG_H would provide a stronger enthalpic driving force to promote the dissociation of water, which might account for the experimental observations. It should be pointed that to describe the reactivity of H_2O dissociation, only considering the adsorption of H was insufficient such that more systematic theoretical studies are needed, which should be done in the future. Our understanding in current stage is too preliminary to be included in the manuscript. We wish the reviewer's understanding.

Figure C1. Different surface sites on: (a) $fcc \{110\}$; (b) $hcp \{11\bar{2}0\}$.

Table C1. Calculated ΔG_H on $fcc \{110\}$ and $hcp \{11\bar{2}0\}$ facets of Ni. (Unit: eV)

Surface	Site	Ni	Pt-Ni alloy	Pt skin
$fcc \{110\}$	Top	0.26	0.30/-0.16	-0.08
	Short bridge	-0.17	-0.22	-0.20
	Long bridge	-0.21	-0.05	-0.08
	Hollow	-0.22	-	-
$hcp \{11\bar{2}0\}$	Top	-	-	-
	Short bridge	-0.19	-0.22	-0.21
	Long bridge	-0.29	-0.18	-0.25
	Hollow	-0.29	-	-

Comments: R2 C3: The authors appears to have misunderstood the comment made, the authors should consider adding literature references/DFT calculations to support the preferred adsorption of oleic acid on the facets that would result in the structure observed, not merely state that it empirically is a required co-solvent (structure directing agent).

Comments: R2 C5: The authors bring up an interesting point here. Have the authors considered if the low coordination of the [110] face of hcp-Ni could favor the binding of structural directing agents such as oleic acid.

Response to R2 C3 and R2 C5: Thank a lot for the reviewer's comment. We carefully surveyed the relevant literatures, it was reported that the oleic acid preferred adsorbing on the side facet of the *hcp* crystal structure, leading to the formation of nanorods or nanowires along the *c*-axis of *hcp* Ni and *hcp* NiCo alloy structure (*Chin. J. Chem.*, 29, 1119 (2011), *J. Mater. Chem.*, 18, 5696 (2008)). The phenomena are very similar to our experimental data. Relevant discussion was added in the revised text (L9-11, Page 6).

Furthermore, following the reviewer's suggestion, we perform DFT calculations to examine the adsorption of oleic acid on *hcp*-Ni {0001} and {11 $\bar{2}$ 0} surfaces. Our DFT calculations demonstrate the adsorptions of oleic acid (OAH) on the {11 $\bar{2}$ 0} surface are favored over those on the {0001} by ~ 0.25 eV, either the molecular adsorption or dissociated adsorption (Supplementary Table 2). This finding indicates that {11 $\bar{2}$ 0} surface could be stabilized by the adsorption of OAH, being consistent with the experimental observations. The revised discussion was highlighted in the text (L13-16, Page 6).

Comments: R2 C6 L. 257 The authors state: "However, the exchange current density of the hcp Pt-Ni alloy ($1.65 \text{ mA}\cdot\text{cm}^{-2}$) is much larger than that of the fcc counterpart and commercial Pt/C ($0.68 \text{ mA}\cdot\text{cm}^{-2}$ and $0.46 \text{ mA}\cdot\text{cm}^{-2}$, respectively)." The authors should compare the exchange current density on the basis of the ECSA not only the geometric surface area. As the j_0 is proportionally dependent on surface area this will allow for a more direct comparison of intrinsic activity. The discussion of the Tafel slope is very useful, the authors should compare the effect of adding Ni to the studies by Markovic (DOI doi:10.1038/nchem.1574) in which alkaline HER is thoroughly discussed.

Response: Thanks for the reviewer's comment. We calculated the exchange current density of different catalysts on the basis of the ECSA (as shown in Supplementary Table 4), and found that the exchange current density of the *hcp* Pt-Ni excavated multipods ($1.65 \text{ mA}\cdot\text{cm}^{-2}$) is much larger than that of the *fcc* counterpart and

commercial Pt/C ($0.68 \text{ mA}\cdot\text{cm}^{-2}$ and $0.46 \text{ mA}\cdot\text{cm}^{-2}$, respectively). In addition, the current density of different catalysts normalized to the geometric surface area, ECSA and the mass of Pt on the electrode are respectively shown in Supplementary Table 4. We can find that the unique *hcp* Pt-Ni excavated multipods exhibit the best catalytic property.

By fitting the experiment data with the *Butler-Volmer* equation, we calculated the *Tafel* slope of the different catalysts (Figure 5c). The *Tafel* slope of the *hcp* Pt-Ni excavated multipods and *fcc* counterpart are $78 \text{ mV}\cdot\text{dec}^{-1}$ and $74 \text{ mV}\cdot\text{dec}^{-1}$, respectively, which were less than that of the commercial Pt/C ($117 \text{ mV}\cdot\text{dec}^{-1}$). The lower *Tafel* slope of the Pt-Ni alloy suggested that the alloying of Ni to Pt facilitates the HER in the alkaline solution. *Markovic* and the coworkers found $\text{Pt}_{0.1}\text{Ru}_{0.9}$ exhibits very high activity for the HOR/HER in alkaline environment due to the more oxophilic sites on Ru atoms which facilitate the adsorption of hydroxyl species (OH_{ad}) (DOI:10.1038/nchem.1574). It is reasonable for us to propose that the alloying Ni atoms are more likely to provide the sites for the adsorption of hydroxyl species comparing with Pt atoms, and consequently, improve the water dissociation step of the alkaline HER activities for Pt-Ni materials. The relevant discussion was revised and the references were added in the manuscript (L7-11, Page 9).